# Land Use Changes Influence the Soil Enzymatic Activity and Nutrient Status in the Polluted Taojia River Basin in Sub-Tropical China

**DOI:** 10.3390/ijerph192113999

**Published:** 2022-10-27

**Authors:** Chenglin Yuan, Siqi Liang, Xiaohong Wu, Taimoor Hassan Farooq, Tingting Liu, Yu Hu, Guangjun Wang, Jun Wang, Wende Yan

**Affiliations:** 1National Engineering Laboratory for Applied Technology in Forestry and Ecology in South China, Central South University of Forestry and Technology, Changsha 410004, China; 2Bangor College China, a Joint School between Bangor University and Central South University of Forestry and Technology, Changsha 410004, China

**Keywords:** Taojia River basin, land use, soil properties

## Abstract

Different land use practices may improve soil quality or lead to soil deterioration. Recently, environmental problems, such as heavy pollution and soil erosion, have led to serious land degradation in the Taojia River basin. In this study, we explored the soil fertility characteristics (mechanical composition; pH; soil organic matter (SOM); soil total nitrogen (TN); and the activity of four enzymes, i.e., urease, hydrogen peroxide, alkaline phosphatase, and sucrose enzymes) under different types of land use in the Taojia River basin. Soil samples were taken from 0–10 cm, 10–20 cm, and 20–40 cm depths from four different land use types that were widely used in the Taojia river basin, including cultivated land, vegetable fields, woodlands, and wastelands. The results showed that the soil enzyme activity and the constituents of the soil were closely related and significantly affected each other (*p* < 0.05). Woodland soil exhibited the highest content of SOM in all soil depths. Soil total nitrogen mainly depended on the accumulation of biomass and the decomposition intensity of organic matter, so the changes in TN followed the trends of the changes in SOM. Woodland soil showed an improved mechanical composition. We were also able to observe an increased clay content in woodland soil. Woodland soil also exhibited the reversal of soil desertification and an increase in nutrient/water retention capacity. Therefore, an increase in woodland areas would be an appropriate goal in terms of land use in order to improve the eco-environmental quality of the Taojia River basin.

## 1. Introduction

Rapid economic growth, urbanization, and intensive resource development have offset the gains arising from ecological conservation, and changes in agricultural land use can be associated with this phenomenon. Agricultural activities are responsible for the greatest ecological damage that humans have created in Hunan Province, with decreases in ecological areas of 229.82 km^2^ and 132.12 km^2^, respectively, during 2000–2010 [1]. Rapid development in urban areas has turned Hunan into one of the provinces that has undergone relatively serious soil and water losses in southern China [2]. It is well known that there are abundant reserves of non-ferrous metals in Hunan Province [3]. Wastewater, waste gas, and other toxic and harmful substances from mining have caused environmental pollution in the surrounding areas, thus resulting in severe ecological damage [4,5]. For example, Xiangtan is a prefecture-level city in Hunan that has been confronted with serious problems, including environmental pollution, the rehabilitation of vegetation, and reductions in crop yield, because of the large amount of mercury, cadmium, and lead contained in the wastewater discharged from industrial smelters. This has resulted in a significant threat to the city and the lives and property of residents in the basin [6]. Changes in land use can be enabled by changes in land cover [7,8,9] or agricultural activities, and this can have significant and lasting effects on the physicochemical and biological properties of the soil [10]. It also plays a vital role in regional and local climate conditions globally. To date, a large number of studies have been carried out on the impacts of soil organic carbon (SOC) and the nutrient characteristics of soil [11,12], whereas fewer studies have been conducted on the relationship between the microbial properties of soil and enzymatic activities [13,14,15]. The activity of soil enzymes greatly influences nutrient cycling, and it plays an important role in sustainable development in agriculture [16].

In a study of the hilly region of the Loess Plateau, the authors argued that artificial reclamation could be a valuable solution for the control of soil erosion and the improvement of eco-environmental conditions in the gully areas of the Loess Plateau [17]. Xiao S.Z. (2021) determined that the total potassium, the total phosphorus, and the phosphorus available in the soil in limestone shrubland were higher than those in the dolomite shrubland. As for the four types of land use most heavily affected by human activities (namely, paddy fields, dryland, flue-cured tobacco fields, and pear orchards), the upper soil layer was found to contain contents of all nutrients (except potassium) that were higher than those in the lower layer [18]. Yan et al. (2021) reported that different types of land use varied significantly in terms of the thicknesses of the soil layers in the following sequence: farmland (42.63 cm) > woodland (39.26 cm) > grassland (20.35 cm) [19]. Kebebew, S. et al. (2022) found that land use change was one of the main reasons for the decline in soil fertility and agricultural productivity in south central Ethiopia. Furthermore, land use changes showed significant differences (*p* < 0.05) in terms of soil OC and total N in a comparison of land use types [20]. These different land use practices may improve soil quality or lead to land degradation, resulting in changes in the enzyme activity of the soil [21].

The Taojia River is located in Chenzhou City, Hunan Province, with an extent of about 58.5 km. It flows through Linwu County, Jiahe County, and Guiyang County, covering a total basin area of 602 km^2^. Woodland and cultivated land are the top two land uses in this area. The Taojia River basin is facing problems of severe heavy metal pollution, soil and water losses, and excessive amounts of suspended matter in the river that have led to sedimentation in the stream [22]. However, there are few studies available on soil enzymes in association with different land use practices in the areas surrounding polluted river basins [23,24]. In this study, we focused on this urgent topic with the aim of speeding up the ecological restoration and rehabilitation in the area and improving the local ecological environment. Therefore, the objectives of this research included: (i) examining the variability in soil quality characteristics, such as soil pH, organic matter content (SOM), total nitrogen (TN), and the activity of soil enzymes (urease, hydrogen peroxide, alkaline phosphatase, and sucrose enzymes), and (ii) identifying the relationship between the activity of soil enzymes and the physiochemical properties of the soil under different land use types and at various depths.

## 2. Study Area and Methods

### 2.1. Location of the Study Area

The study was conducted in Xinmadi Village, Fangyuan Town, Guiyang County, Hunan Province (112°13′–112°55′ E, 25°27′–13°6′ N) (Figure 1). The region has a subtropical monsoon climate with distinct seasons. Precipitation is abundant, with an average annual rainfall of about 1490 mm. The average annual temperature is around 18 °C [25]. The types of soil mainly include red, yellow, and yellow-brown soil [26]. The zonal vegetation type in this subtropical region is evergreen broad-leaved forest. The plant communities are distributed along elevation gradients, including evergreen broadleaved forest below 600 m, evergreen and mixed deciduous forest from 650 m to 1000 m, deciduous hardwood forest from 1000 m to 1500 m, and shrub grass above 1500 m.

### 2.2. Sampling

According to the principle of maintaining consistency between the experimental sample site and the surrounding environment, within the study area, sampling was performed on cultivated land, vegetable farmland, and woodland, as well as in the wastewater area of the Taojia River basin to ensure a similar slope and a consistent slope direction. Different sampling points were selected from locations with similar gradients in the river basin. Three points were chosen for each type, and sampling was carried out repeatedly. Soil samples were taken from various depths between 0–10 cm, 10–20 cm, and 20–30 cm. When sampling was completed, ground-cover vegetation, fallen leaves, gravel, and other miscellaneous materials were removed. A total of 36 samples were collected to analyze the effects of different land use types on the mechanical composition of the soil, pH values, contents of organic matter, total nitrogen, and enzyme activity.

### 2.3. Determination of the Soil Parameters

Parameters, including pH values, contents of organic matter, and total nitrogen of the soil, were measured [27] using the glass electrode method, a laser particle size analyzer, the potassium dichromate volumetric method, and the Kjeldahl nitrogen method, respectively. The activity of soil enzymes, namely, sucrose, urease, alkaline phosphatase, and hydrogen peroxide enzymes, was determined using the dinitro salicylic acid method, the sodium-phenol colorimetric method, the disodium phenyl phosphate method, and the photometric method, respectively [28,29]. The index determination instruments mentioned in this paper include a PHS-3E, an enzyme calibrator, an automatic Kjeldahl nitrogen tester, a microwave digestion instrument, etc.

### 2.4. Data Processing

Analysis of variance (ANOVA) was performed to analyze the differences between the studied parameters under different land use conditions and the soil depth profiles. Moreover, the Pearson correlation test was used to observe the relationships between different soil fertility characteristics. All the values were declared significant at *p* < 0.05. Means that exhibited significant differences were compared using the LSD significance test. All the tests were performed using SPSS [30].

## 3. Results

### 3.1. Effects of Different Land Use Practices on the Physical Properties of Soil

The soils’ textures were classified and divided into sand, silt, and clay, according to the percentage of these components in the soil. The particle size of sand was defined as 0.02–2 mm, while that of silt was defined as 0.002–0.02 mm, and that of clay was below 0.002 mm. In the soil layer between depths of 0–10 cm, the sand content varied among different land use types as follows, in descending order: vegetable field > cultivated land > wasteland > woodland. The silt content varied as follows, in descending order: woodland > cultivated land > wasteland > vegetable field. The clay content varied as follows, in descending order: woodland > wasteland > cultivated land > vegetable field. In the soil layer at depths between 10–20 cm, the sand content varied as follows, in descending order: vegetable field > cultivated land > wasteland > woodland; for silt content: woodland > wasteland > cultivated land > vegetable field; and for clay content: woodland > wasteland > vegetable field > cultivated land. In the soil layer at depths between 20–30 cm, the sand content varied as follows, in descending order: wasteland > woodland > vegetable field > cultivated land, whereas silt and clay exhibited the following identical order of variation: cultivated land > vegetable field > woodland > woodland > wasteland (Figure 2).

### 3.2. Effect of Various Land Use Types on the Chemical Properties of Soil

At depths between 0–10 cm, the pH values varied for different land use types as follows, in descending order: vegetable field > cultivated land > wasteland > woodland (Figure 3a). At depths between 10–20 cm, the pH values varied as follows, in descending order: cultivated land > vegetable filed > wasteland > woodland (Figure 3a). At depths between 20–30 cm, the pH values varied as follows, in descending order: cultivated land > vegetable field > wasteland > woodland (Figure 3a). The organic matter and total nitrogen in the soil at depths between 0–10 cm and 10–20 cm varied as follows, in descending order: cultivated land > vegetable field > woodland > wasteland (Figure 3b,c). At depths between 20–30 cm, the organic matter and total nitrogen in the soil varied as follows, in descending order: woodland > vegetable field > cultivated land > wasteland (Figure 3b,c). Woodland soil had a higher content of organic matter at all depths from 0 to 30 cm. Thus, on balance, woodland showed the best physical and chemical properties. Furthermore, total nitrogen mainly depended on the accumulation of biomass and the decomposition intensity of the organic matter, so the change in total nitrogen followed the trend of the change in the organic matter in the soil.

### 3.3. Changes in Soil Enzyme Activity under Different Land Use Types

From the perspective of different soil layers at depths between 0–10 cm, the content of urease under different land use types decreased as follows: woodland > vegetable field > cultivated land > wasteland (Figure 4a). The content of hydrogen peroxide enzymes decreased as follows: vegetable field > cultivated land > wasteland > woodland (Figure 4b). The content of alkaline phosphatase decreased as follows: vegetable field > woodland > wasteland > cultivated land (Figure 4c). The content of sucrase decreased as follows: vegetable field > cultivated land > woodland > wasteland (Figure 4d). At depths between 10–20 cm, the content of urease decreased as follows: vegetable field > cultivated land > woodland > wasteland (Figure 4a). The content of hydrogen peroxide enzyme decreased as follows: wasteland > vegetable field > cultivated land > woodland (Figure 4b). The content of alkaline phosphatase decreased as follows: woodland > vegetable field > cultivated land > wasteland (Figure 4c). The content of sucrase decreased as follows: vegetable field > waste land > cultivated land > woodland (Figure 4d). At depths between 20–30 cm, the content of urease decreased as follows: vegetable field > wasteland > woodland > cultivated land (Figure 4a). The content of hydrogen peroxide enzymes decreased as follows: cultivated land > vegetable field > wasteland > woodland (Figure 4b). The content of alkaline phosphatase decreased as follows: wasteland > woodland > vegetable field > cultivated land (Figure 4c). The content of sucrase decreased as follows: woodland > vegetable field > woodland > cultivated land (Figure 4d). Concerning the trends regarding the change in enzyme activity, the activity of urease in the soil increased with the sampling depths. The activity of urease in vegetable fields, wasteland, and woodland all decreased, and the activity in cultivated land increased first and then decreased. At deeper sampling points, the activity of hydrogen peroxide enzymes in vegetable fields and cultivated land gradually increased, and the activity in wasteland first increased and then decreased. The activity of hydrogen peroxide enzymes in woodland gradually decreased. The activity of alkaline phosphatase increased at deeper sampling points. The activity of alkaline phosphatase in vegetable fields, cultivated land, and woodland gradually decreased, and the activity in wasteland first decreased and then increased. The activity of sucrase increased at deeper sampling points. It decreased in vegetable fields, cultivated land, and woodland. In wasteland, it first decreased and then increased.

### 3.4. Correlation Analysis of Soil Enzyme Activity, Soil pH, and Soil Nutrients

The correlation analysis of soil enzyme activity, soil pH, and major nutrients showed the following: urease exhibited a significant positive correlation with SOC and a highly significant positive correlation with TN in the soil; catalase exhibited a highly significant positive correlation with the pH value of the soil; alkaline phosphatase exhibited a highly significant positive correlation with total nitrogen in the soil; sucrase exhibited a highly significant positive correlation with total nitrogen in the soil; sucrase exhibited a highly significant positive correlation with the pH value of the soil; and alkaline phosphatase exhibited a highly significant positive correlation with TN. These results indicate that enzyme activity could be an important indicator of changes in soil quality.

The analysis of the enzymatic activity of four soil enzymes, namely, urease, catalase, alkaline phosphatase, and sucrase, showed a significant positive correlation between urease and alkaline phosphatase (*p* < 0.05), a highly significant positive correlation between urease and sucrase (*p* < 0.01), and a highly significant positive correlation between alkaline phosphatase and sucrase (*p* < 0.01) (Figure 5). These correlations indicate that urease activity relates to and interacts with alkaline phosphatase activity in the soil and that sucrase activity, urease activity, and alkaline phosphatase activity relate to and interact with one another.

### 3.5. Correlation Analysis of Physical and Chemical Properties of Soil under Different Land Use Types

The content of clay in the soil had a significant positive correlation with the content of silt, a significant negative correlation with the content of sand, and a significant positive correlation with the pH value. The content of silt had a significant negative correlation with sand, a significant positive correlation with the pH value, a significant positive correlation with the content of the organic matter in the soil, as well as a significant positive correlation with the content of total nitrogen in the soil (Figure 6). The content of total nitrogen had a significant positive correlation with the content of total phosphorus in the soil.

## 4. Discussion

At the depth between 20–30 cm, the particle size of soil in cultivated land was smaller than that in woodland. This result is similar to the findings of Yang Ting et al. 2016, who reported that in cultivated land with a low slope in the loess hilly region, the soil particles were finer than those in woodland [31]. It also corresponds to Mendes’s findings in his study on the quality of red soil, namely, that land reclamation destroyed soil agglomerates [28]. However, at depths between both 0–10 cm and 10–20 cm, woodland exhibited the lowest content of sand. The content of clay was the highest, which differs from the previous findings, probably because the soil type in the Taojia River basin region is different from that in the loess hilly area and different crops grow in these two regions. Soils respond to artificial reclamation differently; therefore, the results of this study are different from the findings of Yang Ting et al. 2016. Likewise, Song Qin 2021 found that in most of the sample plots, the clay content and the silt content in the cultivated land were higher than those in the surrounding uncultivated areas, but some of the sample plots showed declining trends, which indicates that artificial cultivation does not always increase the number of fine particles [29]. Similar results were also obtained in a study by Xia Yu, who reported that the clay and the silt contents in farmland were higher than those in woodland [32]. In all three soil layers, the pH values of wasteland and woodland were significantly different than those of vegetable fields and cultivated land—that is, the pH values of wasteland and woodland were lower. No significant difference between the pH values of moorland and woodland was observed at depths between 0–10 cm and 10–20 cm. The acidity of the soil may be related to artificial fertilization, which changed the pH value of the soil in the region and the accumulation of organic acid in the soil during the decomposition of apoplastic materials by micro-organisms. This result is consistent with Lei Huayang’s finding that the acidity and the alkalinity of soil can affect the formation of clay particles [33]. The contents of organic matter under all four land use types decreased with increasing soil depth, which may be a result of dead leaves and microbial activities gathering at the surface layer of the soil [34]. The content of the organic matter in cultivated soil was higher than that in woodland soils between the depth of 0–20 cm, which is inconsistent with the finding of Yang, Shuyuan et al. 2021 that the content of organic matter in woodland soil was higher than that in cultivated soil at the depth between 0–40 cm [35]. It is probably the pollution caused by heavy metals, including mercury, cadmium, and lead, in the Taojia River watershed that affects the process of the accumulation of organic matter from micro-organisms and dead leaves in the soil [23]. At a depth between 20–30 cm, the total nitrogen content of woodland differed greatly from that of wasteland. At a depth between 0–20 cm, the total nitrogen content of arable soil was the highest among all four land use types, and the total nitrogen content of woodland soil at a depth between 20–30 cm was higher than that of vegetable field, wasteland, and arable soils. This may be because the roots are deeper in woodland and nitrogen fixation can also occur in deeper soil [36]. In addition, the artificial application of fertilizer has resulted in a higher content of total nitrogen in the surface layer of soil in arable land and in vegetable fields compared to woodlands, and the use of fertilizer is also an important factor affecting total nitrogen content in soil in arable land.

## 5. Conclusions

As far as the comprehensive results of this study are concerned, an interaction was observed between urease activity and alkaline phosphatase activity of the soil, along with a close interaction between sucrase activity and urease activity and alkaline phosphatase activity. The results also indicated that in the Taojia River basin, soil enzyme activity and the content of major nutrients in the soil were related to each other. Enzyme activity could be used as an essential indicator of soil quality changes. Overall, woodland soil exhibited the highest levels of major soil nutrients, whereas the observed change in total nitrogen was basically in line with the trends of the organic matter in the soil. Woodland soil showed an improved mechanical composition. In woodland soil, the clay content was increased, and the sand content was reduced. Woodland soil was also capable of reversing soil desertification and increasing the nutrient/water retention capacity. Consequently, an increase in woodland areas would be an appropriate land use practice in the Taojia River basin to improve its ecological environment quality.

## Figures and Tables

**Figure 1 ijerph-19-13999-f001:**
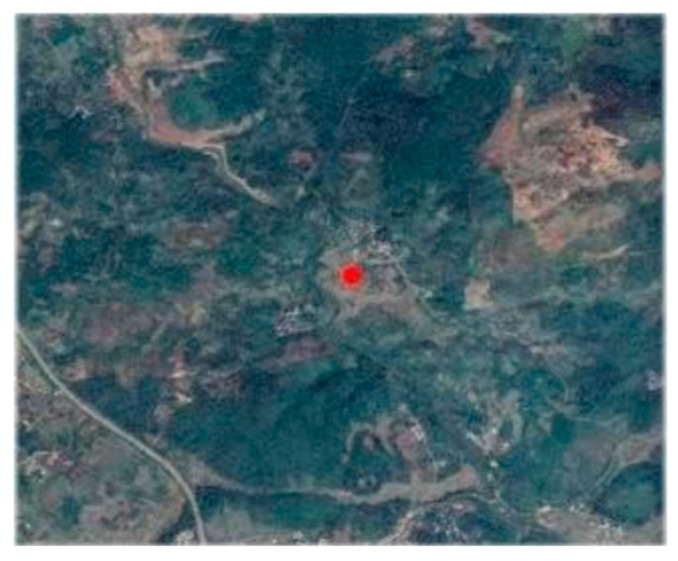
The study area location.

**Figure 2 ijerph-19-13999-f002:**
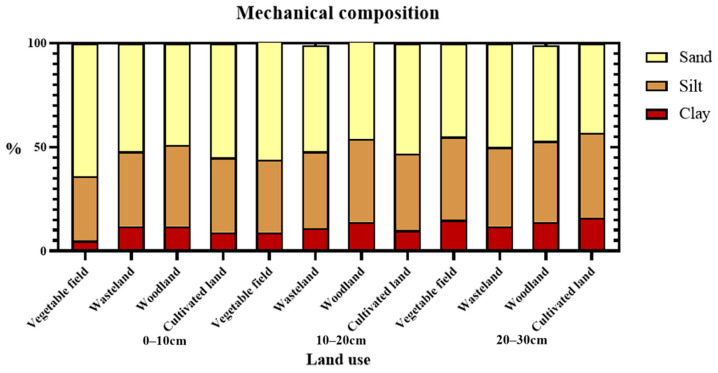
Textures of soils at various depths under different land use types.

**Figure 3 ijerph-19-13999-f003:**
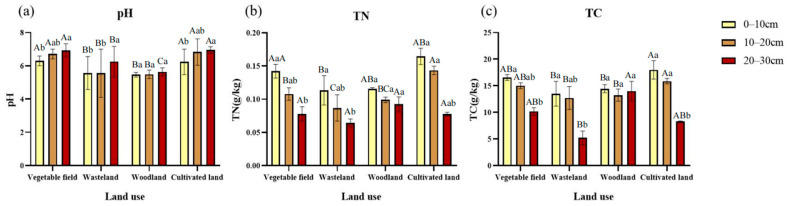
(**a**) Soil pH; (**b**) total nitrogen in soil (TN), and (**c**) organic matter in soil (SOM) for different land use types in the Taojia River basin. The values are means ± SE. Note: The difference is not significant if there is a letter with the same marker, and significant if there is a letter with different markers. Upper case letters indicate the difference between different land uses within the same soil layer. Lower case letters indicate the difference between different soil layers within the same land use pattern.

**Figure 4 ijerph-19-13999-f004:**
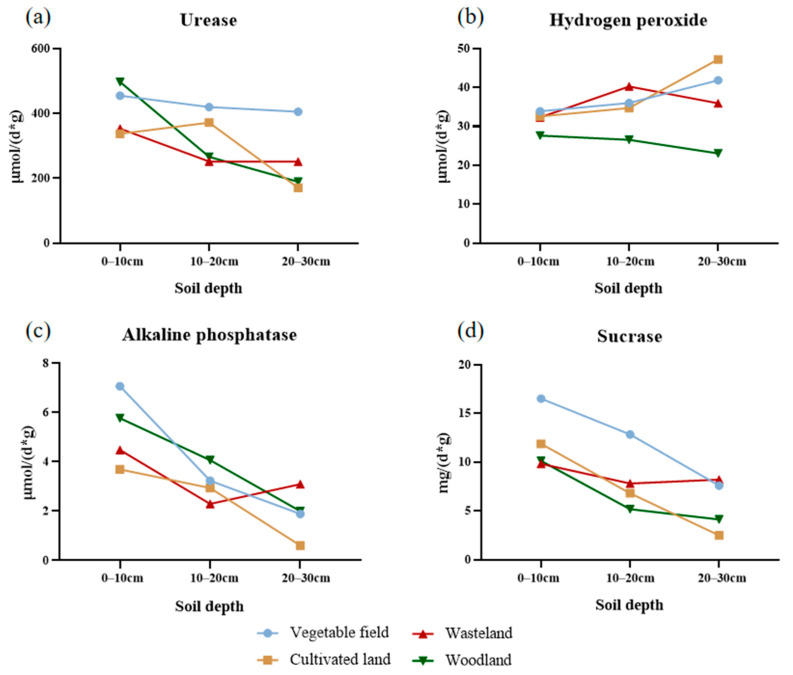
Changes in (**a**) soil urease, (**b**) soil hydrogen peroxide, (**c**) soil alkaline phosphatase, and (**d**) soil sucrase activity under different land use types in the Taojia River basin.

**Figure 5 ijerph-19-13999-f005:**
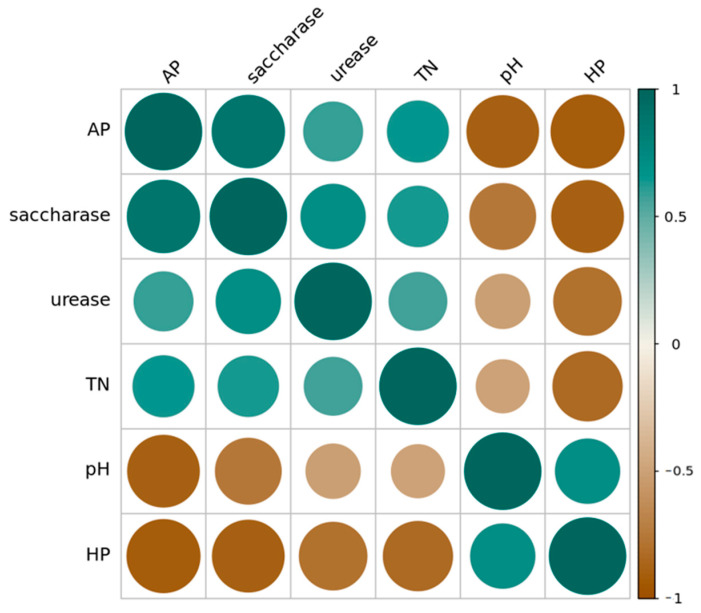
Pearson correlation between soil enzymatic activity, soil pH, and soil nutrients across all land use types and soil depths in the Taojia River basin. Note: brown—darker colors and larger circles indicate *p*-values closer to −1 and a more significant correlation between the two indicators. Green—darker colors and larger circles indicate *p*-values closer to 1 and a more significant correlation between the two indicators.

**Figure 6 ijerph-19-13999-f006:**
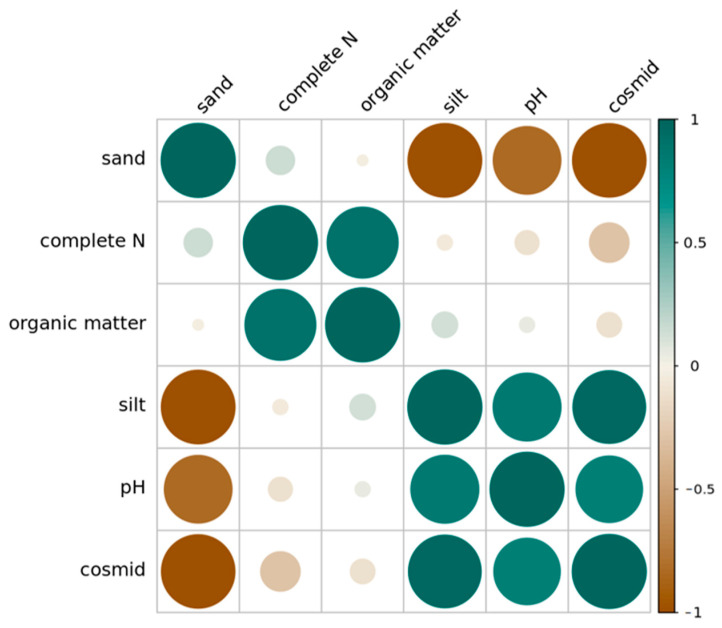
Pearson correlation between soil physical and chemical properties across all land use types and soil depths in the Taojia River basin. Note: brown—darker colors and larger circles indicate *p*-values closer to −1 and a more significant correlation between the two indicators. Green—darker colors and larger circles indicate *p*-values closer to 1 and a more significant correlation between the two indicators.

## Data Availability

The data presented in this study are available on request from the corresponding author. The data are not publicly available due to the funded projects have not been completed.

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
