# Peer review of "Land Use Changes Influence the Soil Enzymatic Activity and Nutrient Status in the Polluted Taojia River Basin in Sub-Tropical China"

_ijerph, 2022, doi:10.3390/ijerph192113999_

Round 1
Reviewer 1 Report
This study explored the soil fertility characteristics and activity of four enzymes under different land use types. Finally, the authors suggested that woodland would be an appropriate way of land use to improve the eco-environmental quality in the study area. Generally speaking, this work is intriguing. This manuscript needs some revise as follows:
Maybe the summary section should be provided some quantitative descriptions, such as major enzyme activities and nutrient content.
Line 109: Firstly, raplace “Aonova”. The correct statement should beYou said “Anova”. “Two-way analysis of variance (Aonova) was perfoemed” in “2.4. Data Processing ”, but there are no pictures or tables show me the results of Two-way anova. If this work did not used it, then the unreasonable claims should be deleted.
Line 148-149: “Fig. 2” needs to add significance labels in the three picture.
Line 163: “At Depths of”. “Depths” should use the lower letters “depths”
Line 225: “ It is because Soils”. Same mistake with line 163.
Reviewer 2 Report
Some comments to improve the quality of the manuscript are given directly on the pdf version of the manuscript (see the attached pdf)

Reviewer 3 Report
The paper "Land Use Changes influence the Soil Enzymatic Activity and Nutrient Status in the Polluted Taojia River Basin in Sub-tropical China" focus on the variability of soil quality characteristics and the relationships between the activity of soil enzymes and physiochemical properties of soil under different land use types. The results are comprehensive and the conclusion is reasonable. I think the manuscript can be accepted after a minor revision. Comments are listed as follows:
1. Advise providing a location map of the study site.
2. In the part of research methods, it is suggested to provide information such as the instrument and model used for the determination of soil chemical and enzyme activity indexes.
3. Line 19, “0-10 cm, 10-20 cm”
4. Line 63, please replace “soil deterioration” using “land degradation”.
5. Line 119, “0-10cm” should be changed to “0-10 cm”. The same problems need to be revised throughout the manuscript.
6. Line 195-197, please change ‘p’ to ‘p’.
7. Please check the references carefully based on the requirement of this journal.
